# Physical Activity and Public Health among People with Disabilities: Research Gaps and Recommendations

**DOI:** 10.3390/ijerph191610436

**Published:** 2022-08-22

**Authors:** Gregory W. Heath, David Levine

**Affiliations:** 1Public Health Program, Department of Health and Human Performance, University of Tennessee, Chattanooga, TN 37403, USA; 2University of Tennessee Health Science Center College of Medicine Chattanooga, Chattanooga, TN 37403, USA; 3Department of Physical Therapy, The University of Tennessee, Chattanooga, TN 37403, USA

**Keywords:** exercise, epidemiology, surveillance, non-communicable diseases, measurement, interventions

## Abstract

Physical activity has become an integral component of public health systems modeling the public health core functions of assessment, policy development, and assurance. However, people with disabilities have often not been included in public health efforts to assess, develop policies, or evaluate the impact of physical activity interventions to promote health and prevent disease among people with disabilities. Addressing the core function of assessment, current physical activity epidemiology, and surveillance among people with disabilities across the globe highlights the paucity of surveillance systems that include physical activity estimates among people with disabilities. The status of valid and reliable physical activity measures among people with condition-specific disabilities is explored, including self-report measures along with wearable devices, and deficiencies in measurement of physical activity. The core functions of policy development and assurance are described in the context of community-based intervention strategies to promote physical activity among people with disabilities. The identification of research gaps in health behavior change, policy, and environmental approaches to promoting physical activity among people with disabilities is explored, along with recommendations based on the principles of inclusive and engaged research partnerships between investigators and the members of the disability community.

## 1. Epidemiology and Surveillance of Physical Activity among People with Disabilities (PWD)

Background and rationale. Epidemiology and public health surveillance are a cornerstone of public health practice used to systematically monitor the population-level trends in health and behaviors, and to guide intervention priorities [1]. Existing surveillance systems for physical activity most often assess the aerobic physical activity (PA) behaviors of individuals and have been used to monitor secular changes in the population levels of PA over time [2,3]. Surveillance can also be used to monitor the factors that can influence changes in the PA levels. For example, the systems can be used to monitor supports for PA within the community and within various settings, such as workplaces. While PA surveillance systems, such as those existing in the United States and other high income countries (HIC) were primarily developed for national and state/region-specific monitoring of PA and inactivity among adults, their use can also be applied regionally and locally [4]. Sometimes the states/regions, but most often local, application of such data not only includes PA monitoring, but also intervention planning and program evaluation [5,6,7]. The expert panel results from a recent workshop conducted by the U.S. Centers for Disease Control and Prevention (CDC) sought to outline the current and future efforts within public health surveillance systems that need to be implemented in a timely, valid, and reliable manner to assess population-wide levels of PA and sedentary behaviors [8]. As examples, the national surveillance systems [2,9,10,11] along with the Behavioral Risk Factor Surveillance System (BRFSS) can be used to generate regional and even sometimes locally specific PA estimates [12]. In the case of the BRFSS, this system has been adapted for local applications by over-sampling in smaller geographic areas to generate locale-specific estimates of PA and related health behaviors [4,7,12]. The local public health practitioners can also use additional monitoring tools, such as community resource inventories [4] and environmental audits, to assess the PA needs and correlates of activity [13,14]. The objective measurements of PA have been demonstrated to be feasible surveillance and evaluation tools. Direct observation [15], accelerometers [16], and pedometers [17] have proven to be feasible, reliable, and valid measures of PA. When comparing these methods of PA assessment, each provides some advantages and disadvantages. In the case of direct observation, this is where trained observers, unobtrusively, record the number of people and their activity (from sedentary through vigorous) observed in various settings, e.g., parks, paths, playgrounds, and sidewalks. This method is valid and reliable (repeatability), but is labor intensive and requires the training of the observers—so is subject to inter-observer error [15]. Wearable devices, such as pedometers and accelerometers, have also demonstrated a good reliability and validity [8]. Pedometers measure the number of steps and need to be calibrated in accordance with the individual’s walking stride length. Accelerometers are more sophisticated measurement tools, and can measure general movement as well as electronically recording periods of sedentariness as well as movement [8]. The movement data from these wearables can be electronically downloaded into data spreadsheets. The cost of these wearables, unlike pedometers, is more expensive and labor intensive for both the subject wearing the device, often for 7 consecutive days, as well as the investigator. The use of such PA measures among PWD has been limited, since, in the case of wearables, the mode of activity is often walking/running and movement of the lower extremities, excluding use among PWD who have mobility impairments. Hence, further efforts to refine the use of accelerometers among PWD using upper extremities and/or trunk movements are currently being explored by investigators.

Finally, the information from the vital statistics and administrative data, such as hospital admission and discharge data, can be used to support the PA and health link [18]. The efforts to improve public health surveillance for PA can incorporate the same principles that are used to improve other public health surveillance efforts. The public health surveillance systems should have a clear purpose and are evaluated on their usefulness (contribution to prevention and control of disease) and their attributes. These attributes include simplicity, flexibility, data quality, acceptability, sensitivity, representativeness, timeliness, and stability (Table 1) [19]. 

Status of global surveillance of PA among country-specific populations. Much has been accomplished over the past two decades regarding global efforts to measure physical activity prevalence and the impact of physical inactivity on health outcomes and mortality. Most of the current updates have been reported in The Lancet’s Global Health Physical Activity Series (LPAS 1, 2, and 3) in 2012, 2016, and 2021 [21,22,23,24,25,26,27,28,29]. A broader and more in-depth overview of the prevalence of physical activity worldwide is available through the Global Observatory for Physical Activity (GoPA!), and the release of their recent 2nd edition *Global Physical Activity Almanac* (Figure 1) [30].

Current Status of global PA surveillance among PWD. In accordance with the recent review of Martin Ginis et al. [31], the following definitions have been used in this review, from studies of surveillance and assessment of PA among people with disabilities: (1) *Disability,* defined in light of the World Health Organization’s (WHO) International Classification of Functioning, Disability and Health (ICF) [32] and in accordance with the WHO Global Disability Plan 2014–2021: Better Health for All People with Disability [33], where estimates measure the domains of health and functioning, and/or domains and core questions of the Washington Group [34], which includes assessment among the respondents regarding difficulty with vision, hearing, walking or climbing steps, remembering or concentrating, self-care activities such as washing all over or dressing, communicating (understanding or being understood); (2) *PA and physical inactivity* are defined in accordance with the WHO Global Recommendations on Physical Activity for Health [35] and other national bodies, where the minimal amount of PA necessary to accrue health benefits for adults is 150 min of moderate aerobic PA per week or 75 min of vigorous PA per week and for children/youth, aged 6–18 years, 60 min of moderate to vigorous PA per day [35,36,37,38]. Persons are considered physically inactive if their levels of PA are less than these guidelines. More recently, Public Health England released a rapid evidence review summarizing the general health benefits of PA among adults with a disability [37]. Subsequently, WHO released the First Global Physical Activity and Sedentary Behavior Guidelines for People Living with Disability [35,39]. These guidelines provide evidence that doses of PA, tailored to the type of disability, along with guidelines limiting sedentary behavior provide significant health and improved function among PWD, similar to those benefits experienced by people without disability [35].

The current prevalence estimates of PA among people with disabilities comes almost exclusively from among the High Income Countries (HIC), such as those in North America, Western Europe, and Scandinavia [40,41,42,43]. There are prevalence estimates for PA among people with disabilities from Low to Middle Income Countries (LMIC), but these are limited to a small number of countries [44,45,46,47,48]. For adults living in HIC, the estimates among PWD meeting the WHO Guidelines range from 20.6 to 50%, while for PWD living in LMIC, the range is 23.4 to 50%. For children and adolescents, the estimates of those meeting the PA guidelines in HIC vary from 25 to 42%, while no systematic data exist for children and youth estimates in LMICs [49,50,51,52]. These contrast with PA prevalence estimates among children/adolescents and adults without disabilities in both HIC and LMIC of 40% to 60% and 50% to 80%, respectively [38,39,40,45,47,48,49,53]. The known efforts to measure PA and disability among children, adolescents, and adults concurrently across the globe in both HIC and LMIC are summarized in Table 2. 

Currently, comprehensive global estimates of PA patterns among people with disabilities do not exist [31]. The current international survey instruments and surveillance systems have either good measures for PA and sedentary behavior or good measures for disability and functional health, but seldom both. Finally, the Global Burden of Disease (GBD) reports estimates for PA in association with disability-adjusted life years (DALYS) and years lived with a disability (YLD) for specific ICD10 conditions/diseases/injuries [54]. However, it is unclear whether these measures are sufficiently sensitive for people with disabilities, or whether disability advocates find such measures acceptable. 

Research Gaps and Recommendations. Current gaps in global epidemiology and surveillance of PA patterns—sedentary behavior through to high-intensity PA among people with disabilities—include a paucity of country-specific policies supporting improved access to places for PA among persons with disabilities. The current data come primarily from HIC, with little data from LMIC. The current global PA surveillance systems (e.g., WHO STEPs) for NCD do not include functioning and disability survey items. Often, disability surveillance instruments lack PA measures. This may be due in part to the lack of consensus regarding both reliable and valid measures of PA among PWD [31]. The national estimates for PA prevalence are the foundation of the PA action plans. They allow countries to track progress, ideally across all of the population groups and at all stages of the life course. The inclusion of PA prevalence estimates among adults and children with disabilities is now more frequent (e.g., USA, UK), reflecting the call by the WHO Global Action Plan on Physical Activity for “equity across the life course”. However, globally, and especially among LMIC, such data rarely exist, despite its obvious importance. Hence, the extent to which people with disabilities are included in the PA surveillance is remiss, because the national and international PA surveillance data across the globe rarely include people with a disability or classifies them as a group worthy of identification. Unless we understand the different levels of current activity and ability, any actions to shift the priority towards addressing disparities and reducing inequalities will remain both unevaluated and unachieved. The application of such strategies for PA surveillance among PWD require several important steps: (1) the continued evolution of more sensitive and specific PA measurement methods designed to capture the essential and functional PA among PWD, including electronic and other methods; (2) the intentional engagement of PWD in the design and development of PA measures that are adaptable and functionally acceptable; (3) the purposeful collaboration of public health organizations/practitioners with disability organizations and advocates to ensure the full integration of PWD into the mainstream of PA and public health. Thus, it is a priority that PA surveillance be inclusive at multiple levels. For example, action at the international level is critically needed to ensure that the WHO’s “STEP-wise approach to noncommunicable disease risk factor surveillance” data include people living with a disability. Specific strategies to improve physical activity surveillance that need expedient attention in general, fall into a series of categories that were generated by the recent CDC convening of PA and public health experts. ^9^ The research priorities were developed for each of the following strategies: 

(1) Identify and prioritize PA constructs; (2) Assess the psychometric properties of instruments for PA surveillance; (3) Provide training and technical assistance for those collecting, analyzing, or interpreting surveillance data; (4) Explore accessing data from alternative sources; (5) Improve communication, translation, and dissemination of information about estimates of PA from surveillance systems. Although these strategies and research priorities were developed for physical activity surveillance in general, they are directly applicable to PWD, as part of an inclusive surveillance effort for PA [31].

## 2. Physical Activity Measures among People with Condition-Specific Disabilities: Research Gaps and Recommendations

Drawing on the systematic and scoping reviews presented by Martin Ginis et al., [31] which address the PA measures and intervention effects among persons living with a disability, a further search of the subsequent literature was conducted through several search engines, including Pubmed, Scopus, and Web of Science, providing both confirmation of the existing, and an update of the most recent, literature. Potentially disabling conditions were prioritized based on: (1) the prevalence of the condition across the global population; (2) the potential disabling impact of the condition over the course of the lifecycle; and (3) an existing evidence-base demonstrating a positive physical and/or mental health benefit among people living with a disability. 

Persons with Spinal Cord Injury. The accurate measurement of PA in individuals with spinal cord injuries (SCI) has inherent limitations. The measures most used have include self-reported outcomes, such as the Physical Activity Recall Assessment for People with Spinal Cord Injury (PARA-SCI) [55], which has been validated for this population [56]. The PARA-SCI is a three-day PA questionnaire guided by an interview [57].

Accelerometry, which is less dependent on the recall of the activity level and intensity, has also been recommended as a measure of PA for individuals with SCI [58,59,60]. Accelerometers were placed on wheelchair spokes in various studies; however, this does not reflect upper extremity activity, and does not reflect the time that the wheelchair may be coasting or going downhill [60,61]. The placement of accelerometers on the upper extremities is therefore recommended, as well as individual calibration of the accelerometers for this population, to improve the accuracy [58,59]. The studies examining the agreement between accelerometers and the PARA-SCI have shown low levels of agreement. For example, resistance activities (upper body strengthening) and pushing a wheelchair uphill or on varied surfaces was reported higher on the PARA-SCI than with an accelerometer. Conversely, brief periods of activity and some of the activities of daily living may not be captured on the PARA-SCI, but are captured by accelerometry. This suggests that multiple measures may need to be utilized to capture a more complete picture of physical activity in this population [60]. Accurately capturing the amount of PA performed in this population is an area of research that will continue to evolve, but multiple measures combined will yield more accurate data.

Persons with stroke. A stroke is a condition/clinical event which can result in a wide variety of disability outcomes, both in terms of the systems affected as well as the severity and presents several significant PA measurement challenges. The focus here is on mobility impairments resulting from a stroke, whether thromboembolic or hemorrhagic. The physical activity levels of people with stroke are lower than their age-matched counterparts, even when they return to living in the community [62]. Regular PA in stroke survivors can improve the strength, balance, and HRQOL [63]. The methods to assess the duration, intensity, and frequency of exercise and stroke survivors, and its relationship to improving HRQOL, are sparse.

A recent systematic review [64] highlighted the lack of research that examines the measurement properties of the self-report PA assessment tools for stroke survivors. The validity and usefulness of these measures need to be examined in much greater depth, to assess if they are useful in research and in clinical practice. 

The quantitative methods of monitoring PA following stroke were reviewed by [62], and examined the various methods and devices used for the measurement of PA. The objective measures included pedometers, actometers, accelerometers, and inertial measurement units (IMUs) which primarily used accelerometer data. The validity of many of the methods was not assessed and the studies that measured the validity ranged from poor to good, with better validity in more controlled environments, such as an in-patient setting where the daily routines are consistent. The reliability was examined to a higher degree, but there are limited data in real world environments. 

The most recent systematic review [65] also examined the agreement between sensor-based measures of PA and clinical outcome measures, such as the National Institute of Health Stroke Scale, the Fugl-Meyer upper extremity motor assessment scale, Stroke Impairment Scale, etc., and reported that the sensor-based measures represent a different construct compared to the clinical scales [65]. 

Similar to other disabilities, no single device or method, such as patient-reported activity, was found to be ideal for quantifying PA after stroke [62]. The use of sensor-based measurements in combination with clinical outcome measures may yield the best information, similar to the other populations with disabilities. Obtaining accurate measures of activity duration, frequency, and intensity needs to be better refined for this population. What constitutes PA also varies between the studies and a more standardized definition of this would enhance the research in this field. The maintenance of PA after research trials also needs to be addressed for this population [66]. A Cochrane review [67] found only four small trials that examined the efficacy of activity monitors for increasing PA after stroke. The evidence for their usefulness in increasing PA to improve health and HRQOL and prevent a subsequent stroke should be a priority.

Persons with Parkinson’s Disease. The individuals with Parkinson’s disease (PD) have decreased levels of PA compared to age-matched controls [68]. This can be explained in part by the motor control issues due to akinesia and bradykinesia, tremor, and rigidity. A fear of falls has also been a factor limiting PA in this population [69]. The methods for the accurate assessment of PA for individuals with PD have been examined, but the data are limited, with studies that address reliability, validity, and clinical usefulness significantly lacking. The functional outcomes, such as recall of activities questionnaires, have been utilized but can also be affected by cognitive deficits [70]. 

The majority of the scales used have focused on the activities of daily living, and the measures of PA have not been developed specifically for this population, rather, the generic PA measures have traditionally been used, such as the Physical Activity Scale for the Elderly, International Physical Activity Questionnaire-Short Form, Exercise Self-Efficacy Scale, and others [71,72]. 

The progression of PD also makes it important to look at PA over time in the same cohorts of individuals, and how it affects HRQOL, including falls and depression. The data acquired from sensors, such as accelerometers, have increased as the cost of these devices has progressively decreased [73,74,75], but lack standardization and links to health-related measures. One study found that involuntary movements may increase the measures of PA [76], and may need to be factored in. There is a clear need to develop objective, reliable and validated tools for the measurement of PA in individuals with PD, as well PA questionnaires specifically for this population.

People with Cerebral Palsy. Cerebral palsy (CP) is a permanent disorder that commonly affects motor function and is frequently accompanied by comorbidities, such as cognitive and learning disabilities, epilepsy, and visual impairments. The functional limitations commonly seen with CP include difficulty with gait, balance, and increased muscle tone, which can create challenges to performing regular PA [77]. A lack of opportunity for regular PA is also problematic, as well as decreased self-efficacy for exercise. This, coupled with communication and behavioral issues in some of the individuals, compounds the problem. The guidelines for exercise and PA in individuals with cerebral palsy (CP) have been reported in the literature [78], however the majority of the literature has examined PA in children, with few studies on adults. Information is also lacking on how the habits developed in childhood may impact future PA habits, and how PA may impact health over the lifespan [78]. A systematic review examining the fitness measures for individuals with CP was conducted, and, of the over 800 articles they identified, less than one percent examined the reliability or validity of cardiovascular or strengthening measures for individuals with CP [79]. The outcomes’ measures that are based on standardized activity diaries and measures have been sparsely utilized in this population, and adaptations to these scales for the individuals with CP are lacking. While the PA questionnaire for older children (PAQ-C) has been used extensively in normal children, no studies have examined its usefulness in children with CP. The pediatric outcomes data-collection instrument is being used in one study [80], however, this is an ongoing trial. Other functional outcome measures, such as the timed-up-and-go and the child and adolescent scale of participation, have been used as a surrogate for physical activity, but have not been validated for this population. The international physical activity questionnaire has been utilized [81,82] and was found to not be a valid tool for the measurement of PA in individuals with CP [81]. The children’s assessment of participation and enjoyment was used to examine participation in leisure activities, and the reliability was examined but not the validity [83], and this scale does not address duration, intensity, dose, or frequency of PA.

Accelerometry has been used to a large extent in this population for ambulatory individuals, and has been shown to be reliable and valid [80,84,85,86] if adapted to the individual and their level of function. The individuals with cerebral palsy (CP) that are non-ambulatory and that require ambulation aids or wheelchairs have increased challenges in reaching the PA requirements. A few studies have examined physical activity in this population [87]. The quality measures to examine physical activity in individuals with CP are severely limited and need to be developed to better examine PA levels, and the effects of PA on HRQOL and non-communicable disease prevention [88]. The heterogeneity of this population makes the assessment of physical activity challenging without individual adaptations, probably contributing to this gap in the research. 

## 3. Community-Based Interventions to Promote Physical Activity among People with Disabilities

Current Status. Martin Ginis et al. [31] recently conducted a systematic review of PA interventions among PWD. Their findings, which included both primary studies as well as other reviews and meta-analyses, found that most of the PA interventions conducted among PWD have addressed increases in leisure-time PA, were conducted primarily in HIC, and focused on intrapersonal factors and/or interpersonal factors [31]. However, their scoping review did identify several PA interventions delivered at community levels. Generally, such intervention strategies seek a change in the knowledge or practices among individuals, organizations, or community settings (e.g., neighborhoods, schools, parks, recreation centers). The examples include developing guidelines for the construction of accessible built and natural environments (e.g., trails, recreation centers, pools, outdoor venues), developing inclusiveness training programs for PA practitioners, and establishing programs that loan equipment for adapted physical activities [31]. The body of evidence for these types of interventions have produced mixed findings [31]. The examples include a nationwide Canadian study that found an educational intervention, designed to strengthen health care providers’ intentions to discuss LTPA with patients with physical disabilities, had no long-term effects [89]. Another example comes from the Netherlands, where staff training was provided to staff among 18 rehabilitation centers on how to deliver PA counselling with referrals made to hospital-based and community-based PA providers [90,91]. This national program reached 5873 patients with various disabilities and had a significant impact on PA participation over a three-year period [90,92]. 

The policy-level interventions include efforts to change legislation, laws, codes, regulations, rules, and practices that are developed and implemented by governments, government agencies, and nongovernmental organizations, such as businesses and schools. Some of the examples include policies to fund sports programs and equipment for PWD, to provide accessible transportation, and to ensure that built environments are accessible [93,94]. While some of the policy-level changes have proved to be effective for increasing PA in the general population [95], a paucity of studies exist that test the effectiveness of policy changes for increasing PA among PWD.

Such population-level interventions could be successful at encouraging/possibly changing behavior in the population, but with the unintended cost of widening the inequality gaps between the most vulnerable subgroups, including PWDs and the rest of the population. Hence, where the targeted approaches may not be feasible due to practical and/or political constraints, blended or so-called proportionate universal approaches should be considered by policy makers (e.g., refurbishing a park, or providing enhanced access to PA opportunities to those more in need).

The institutional-, community-, and policy-level approaches to promoting physical activity typically target entire communities or segments of communities [96]. At the institutional- and community-level, many of the organizations (e.g., schools, recreation centers, not-for-profit organizations) are responsible for delivering programs to increase physical activity participation in both the general population as well as those with disabilities. There is strong evidence that physical activity interventions carried out by various organizations, and that include informational, behavioral, and social strategies, can increase physical activity among people of all ages [25,97,98,99]. Unfortunately, very few studies have evaluated the physical activity and health impact of such community-based interventions among people with disabilities. The work of several groups have systematically reviewed the use and evidence of community-based intervention strategies among PWD, only to find a paucity of studies and those that exist are wanting in evidence of effectiveness [100,101,102,103].

The policy-level approaches represent rules, regulations, and practices that may influence physical activity through a variety of mechanisms, such as changing the built environment, providing incentives for exercising, or developing national or regional physical activity guidelines [104,105]. While there is sufficient evidence in support of various policy-level interventions to increase physical activity in the general population, representing both HIC and LMIC, [106,107,108] very few studies have examined the effects of these interventions on people with disabilities. Furthermore, the targets or moderators of policy-level interventions have not been well studied among the people with disabilities. For example, the policies that aim to increase physical activity by improving access to the pedestrian and bicycle infrastructure, or altering the environmental design and land use, have proven effective among the general population (see Table 3) [105,106,108,109,110]. The effects of such policy changes on the physical activity levels of people with disabilities have not been examined. Based on the descriptive studies, it is unclear whether the pedestrian infrastructure even plays a significant role in the physical activity levels of people with disabilities [109]. 

Nevertheless, across the globe, many of the barriers to physical activity participation for people with disabilities could be alleviated by national/regional public and organizational policies [92,111,112,113,114]. These policies must align with, and target disability-specific influences on physical activity and be part of broader strategies that target multiple levels of influence [94]. The research has shown, for instance, that national policies supporting investments in major sporting events (e.g., the Paralympics and the Commonwealth Games) often have the intended purpose of enhancing the physical and social accessibility of facilities and venues to people with disabilities [102]. However, the benefits of such developments are unequal, poorly distributed, and do little to address the long-term systemic barriers faced by people with disabilities in the urban environment. For the people with disabilities to take advantage of the accessible recreational facilities, organizational and public policies are needed to alleviate transportation and financial barriers; institutional-, community- and interpersonal-level interventions are needed to address the negative societal attitudes toward physical activity for people with disabilities; and intrapersonal-level interventions are required to teach behavior change techniques [31,95].

## 4. Summary and Recommendations Addressing Research Gaps in PA and Public Health Efforts among People with Disabilities

Although much has been accomplished in the past decade to address the needs for PWD regarding the health-promoting benefits of PA, significant research gaps continue to exist. Such gaps currently limit the development of evidence-based approaches for promoting PA among PWD within a public health context. As has been reviewed in the preceding sections, these gaps fall into three major categories: (1) the need for adequate and inclusive measures of PA among children, youth, adults, and older adults who live with a disability across all of the public health surveillance systems—national, regional, and local; (2) the need for valid and reliable measures of PA among PWD specific to the type of disability and across multiple modes of assessment, including wearable devices, self-report measures, use of senser technology, and proxy measures; and (3) the need for research efforts to identify effective community-based PA promotion strategies that are both inclusive and adapted/tailored for PWD across a wide spectrum of disabilities. 

The research recommendations to address these key components of public health efforts to promote PA among PWD have been enumerated within each of the preceding sections of this review. We have drawn heavily from the work of others to highlight the current status and research gaps, that, if filled through the next generation of research studies covering surveillance, PA measures, and community-based strategies to promote the healthful benefits of PA among PWD, will constitute significant health improvements for PWD across the globe. However, such efforts require significant input and partnership with members of the PWD community. These research partnerships have been identified and implemented across a number of research disciplines, and comprise a strategy that lends itself to addressing the research gaps covered within the context of this review [114]. One such research paradigm which has been used successfully to address the research development for PWD is Integrated Knowledge Translation (IKT) [115]. IKT consists of eight consensus-based principles: (1) Partners develop and maintain relationships based on trust, respect, dignity, and transparency; (2) Partners share in decision-making; (3) Partners foster open, honest, and responsive communication; (4) Partners recognize, value, and share their diverse expertise and knowledge; (5) Partners are flexible and receptive in tailoring the research approach to match the aims and context of the project; (6) Partners can meaningfully benefit by participating in the partnership; (7) Partners address ethical considerations; and (8) Partners respect the practical considerations and financial constraints of all of the partners. Although this model of research partnership with the PWD community was initially carried out in addressing the research efforts among people with a spinal cord injury, the success of this approach appears most applicable in providing important guidance in addressing the research gaps that exist for the full integration of PWD into the context of PA and public health [91,115].

## 5. Conclusions

Major gains have been made globally over the past decade in advancing the importance of the inclusion of people living with a disability into the fabric of public health policies, practice, and engagement. However, there remains significant research deficiencies addressing physical activity and public health among PWD. These research gaps include incomplete public health surveillance of PA, a lack of disability-specific PA measurement, and effective community-based promotion of physical activity among PWD. These research gaps can be overcome through an intentional partnership between the PA and public health research community, public health policy makers, and public health/prevention funding agencies/organizations in concert with people who live with a disability.

## Figures and Tables

**Figure 1 ijerph-19-10436-f001:**
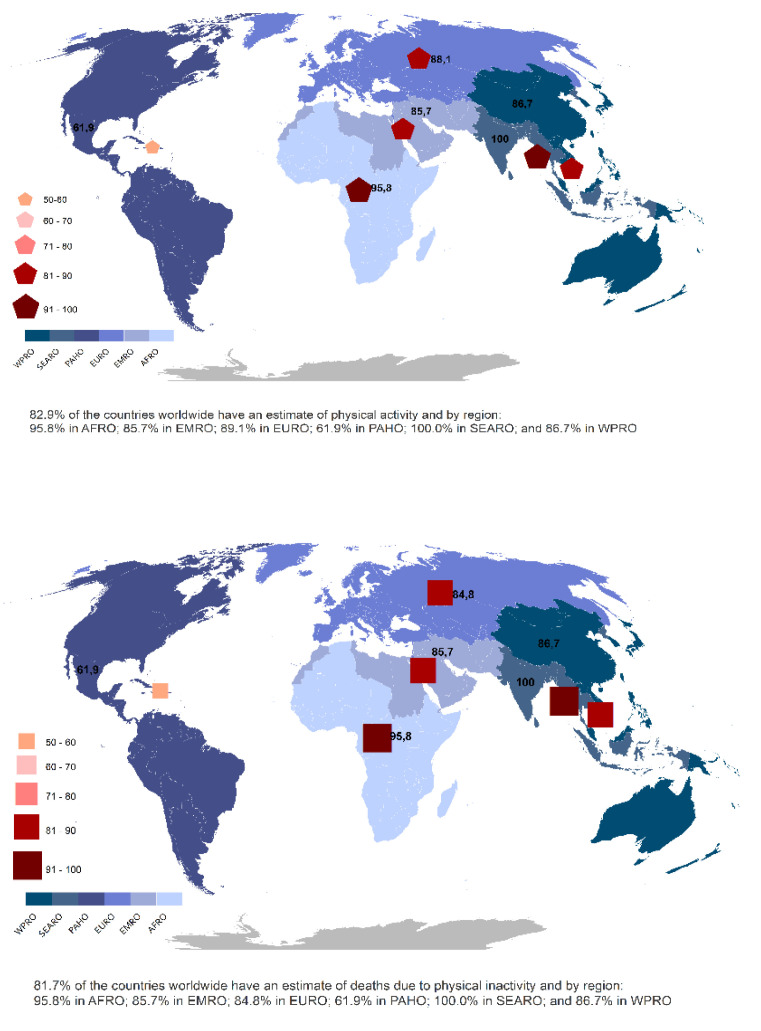
Global prevalence of physical activity and global deaths due to physical inactivity. Adapted with permission from [30].

**Table 1 ijerph-19-10436-t001:** Key attributes to consider when developing or evaluating public health surveillance systems *.

Attribute	Definition	Key Question(s) to Address	Physical Activity Example
Simplicity	Surveillance systems should be as simple as possible while still meeting their objectives.	Are procedures, training, data collection, computer requirements, etc., simple to use?	Training interviewers or those conducting environmental audits does not take excessive time. Physical activity questions are not overly burdensome for the interviewer or the participant.
Flexibility	Can adapt to changing information needs or operating conditions with little additional cost in time, personnel, or allocated funds. Can accommodate, for example, changes in definitions and variations in reporting sources.	Can the system adapt to changing needs and conditions?	Able to adapt to changes in guideline definitions. For example, two definitions of “sufficiently active” (e.g., *Healthy People 2030* and the *2018 Physical Activity Guidelines*) can be assessed in the surveillance system.
Data quality	Completeness and validity of the data recorded in the public health surveillance system.	What proportion of respondents provide incomplete responses to items on the surveillance form?	Respondents can accurately answer a question about their physical activity behavior or community supports.
Acceptability	Reflects the willingness of individuals and organizations to participate in the surveillance system	Will dropout be a factor? Does the system, because of its characteristics, discourage participation?	Participants are willing to complete a questionnaire or wear a monitor for the specified specific period. Employers are willing to complete a tool to assess physical activity supports at their workplace.
Sensitivity	The system should be able to accurately classify participants according to the health outcome of interest and should also be able to detect changes in the prevalence of the health outcome over time (i.e., trends).	Does the instrument accurately classify participants into meeting guidelines or having a community support?	Questionnaires_ENREF_256 [20] have been shown to be valid for assessing community supports for physical activity [20].
Representativeness	Accurately describes the occurrence of a health-related event over time and its distribution in the population by place and person.	Are those who participate (or who are included for environmental systems) different from those who are not?	The proportion of adults meeting guidelines should be similar among those who complete the physical activity interview and among those who do not complete the interview or who are unable to be contacted.
Timeliness	Reflects the speed or delay between the stages of surveillance, such as data collection and reporting. Timeliness evaluated in terms of availability of relevant information—either for immediate efforts or for long-term program planning.	Is time from data collection to dissemination reasonable?	For example, with the BRFSS, time from data collection to having data available for analysis is about 6 months.
Stability	Reliability (i.e., the ability to collect, manage, and provide data properly without failure) and availability (the ability to be operational when it is needed) of the public health surveillance system	Are the tools used for data collection stable over time?	The availability of wearable devices that consistently assess human movement over time.

* Attributes and definitions adapted with permission from [19].

**Table 2 ijerph-19-10436-t002:** Prevalence Estimates of Physical Activity among people with disabilities from around the globe: children; adolescents; adults; and older adults.

Survey Instrument Country of Origin World Bank Economic Strata *	Data Methods Survey Year Population(s)	Physical Activity (PA) Domains **	Disability Domains ***	Prevalence Estimates PA and/or PI (Physical Inactivity)
National Health Interview Survey (NHIS) United States HIC	Face-to-Face Interviews 2009–2012 Adults 15 years and Older	PA and PI	FFL WGQ	PA % (95% CI): FFL-31.0 (29.7–32.2) Vision-45.2 (42.2–48.2); Hearing-40.9 (37.7–44.2); Mobility-38.3 (35.6–41.1); Cognitive-20.6 (19.2–22.1)
National Health and Nutrition Examination Survey (NHANES) United States HIC	Face-to-Face Interviews/Parental surrogates 2011–2014 Children 5–11 years (*n* = 2847)	PA and PI	FFL	PA %: FFL–total = 56.0% males = 58.1% females = 52%
Behavioral Risk Factor Surveillance System (BRFSS) United States HIC	Telephone-based interviewer-led surveys 2009 Adults 18+ years (*n* = 357,665)	PA and PI	FFL	PA % (95% CI): 50% (17.2–32.8)
Active Lives Adult Survey Sport England United Kingdom HIC	Telephone Survey 2016–2017 Adults 18 years and older (*n* ~198,000)	PA and PI	FFL	PA %: Total = 43% 1 impairment = 51% 2 impairments = 45% 3 impairments+ = 36%
Canadian National Longitudinal Survey of Children and Youth (NLSCY) Human Resources and Skills Development Canada (HRSDC) and Statistics Canada HIC	Household Surveys 2006–2007 Children 4–9 years old (*n* 22,431)	PA and PI	FFL—neurodevelopment disabilities affecting mobility	PI % 45%
WHO Collaborative Cross-national Health Behavior in School-aged Children (HBSC) study 15 European Countries HIC/LMIC	School-based Surveys 2013–2014 Adolescents 11, 13, and 15 years (*n* 61,329)	PA and PI	FFL	PA % (prevalence range across all 15 countries): Males = 14.9–37.8% Females = 8.5–21.4%
National Health and Morbidity Survey 2015 Institute for Public Health, Ministry of Health Malaysia LMIC	Face-to-Face Interviews–nationally representative sample 2015 Adults 18 years and older (*n* 19,959)	PI	WGQ	PI % (95% CI): 24.4% (1.24, 1.64)

* High Income Country (HIC); Low to Middle Income Country (LMIC). ** Physical Activity (PA): Adults (18 years and older)–meets WHO and National Guidelines for moderate aerobic physical activity of 150 min per week or 75 min of vigorous aerobic physical activity per week; Children/Adolescents (6 years–18 years)–meets WHO and national guidelines of 60 min of moderate to vigorous aerobic physical activity per week. Physical Inactivity (PI): no reported physical activity + any physical activity less than the WHO guidelines. *** Functioning and Functional limitations (FFL): where functioning and disability represent the interaction between health conditions and (diseases, disorders, and injuries) and contextual factors (external environmental and internal personal factors). Washington Group Questions (WGQ): difficulty functioning in any of the core domains of vision, hearing, mobility, cognition, self-care, and language communication.

**Table 3 ijerph-19-10436-t003:** Overview of strategies to promote physical activity in communities.

Approaches	Strategy	Classification
**Campaigns and Informational**	Point-of-Decision prompts *	Effective
	Community-wide campaigns *	Effective
	Mass media campaigns **	Promising
	Short informational messages **	Promising
**Behavioral and Social**	School-based strategies–physical education *	Effective
	Social Support in communities *	Effective
	Family-based physical activity *	Effective
	Combined diet and physical activity promotion programs to prevent type 2 diabetes among people at increased risk *	Effective
	Physical activity interventions which include activity monitors for overweight or obese adults *	Effective
	Physical activity digital health interventions for adults 55 years and older *	Effective
	Provider-based assessment and counseling **	Promising
	Community physical activity classes **	Promising
**Policy and Environmental**	Physical Activity: Built Environment Approaches Combining Transportation System Interventions with Land Use and Environmental Design *	Effective
	Creating or improving places for physical activity *	Effective
	Interventions to increase active travel to school *	Effective
	Community-wide planning and policies **	Promising

* Adapted with permission from [110]. ** Adapted with permission from [98].

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
