# Peer review of "Physical Activity and Public Health among People with Disabilities: Research Gaps and Recommendations"

_ijerph, 2022, doi:10.3390/ijerph191610436_

Round 1

Reviewer 1 Report

Thank you for the opportunity to review this manuscript. I believe this article addresses an important topic which has been nicely described. I have only a few relatively minor / editorial comments:

- lines 56-57 'Direct observation[15], accelerometers [16] and pedometers [17] have proven to be feasible, reliable, and valid measures of physical activity'. I'd like to see a brief section being dedicated here to that definition / description of these measures (for the lay audience), where the validity of these measures is briefly compared with one another (I'd argue that direct observation is not as reliable as accelerometers, but sometimes it is the only available solution).

- line 84 physical activity abbreviation "PA used only now, it should come earlier in the text

- lines 167-169: although I take the point that those strategies are directly applicable to PWDs, given the manuscript focus, I'd dedicate a couple of lines reflecting on what challenges (methodological / practical) of doing so would be (e.g. outreach?)

- line 176 please remove 'have'

-paragraph starting at line 195 'Stroke' - I would acknowledge that this is a condition / clinical event which can result in a wide variety of disability statuses, both in terms of location and severity

- the authors could explain the reasons for choosing those chronic conditions related to disability at Section 2 - is it due to highest prevalence or just select examples? (to make the argument manageable within one manuscript?)

- paragraph starting at line 328 - this is an important paragraph. The authors could mention the fact the population-level interventions could be successful at encouraging / possibly changing behavior in the population, but at the cost of widening the inequality gaps between the most vulnerable subgroups (e.g. PWDs) and the rest of the population. Where targeted approaches will not be feasible due to practical / political constraints, blended or also called proportionate universal approaches should be considered by policy makers (e.g. refurbishing a park + providing enhanced access to PA opportunities to those more in need).

Reviewer 2 Report

General comments

In this narrative review, the authors described the state of the art regarding physical activity and public health among people with disabilities, discussing research gaps and providing recommendations.

I have no particular suggestions to give to the authors because it is a narrative review that clearly describes the state of the art and presents an adequate number of bibliographical references.

I just have a few comments to make.

Specific comments

Lines 28, 173, 286, 362:  Remove the word “Section”

Line 26: Remove the keyword “physical activity; research gaps; disabilities”. To optimize the search for the manuscript through search engines, I recommend that you add keywords other than those of the title.

Reviewer 3 Report

Thank you for the opportunity to review this paper. Evidence supporting the use of PA in PWD is lacking, has many gaps and is inconclusive in its findings. These types of papers are important to illustrate this and encourage better quality research in this area.

Major concerns:

My primary concern that relates to my comment about ethics is that as a review paper there is no evidence of a systematic search to identify the papers that were used to produce the review. While I understand it is not a SR, there seems to be a large number of self-citations for the first author which would be much easier to justify if the search criteria/strategy to identify these papers is described. Otherwise it appears to be a review of a large number of papers by the first author (7 papers?). If this can be included, then the issue related to self-citation would be removed (as it would be clearly evidenced that they were appropriately selected). This would also help to ensure that populations across more diverse backgrounds will be identified if the data exists.

The second major fault in the paper is the reference list. It is completely incorrect from the first to the last paper. Initials from 2nd author on are in the wrong place, none of the proper nouns are capitalised, none of the acronyms are capitalised, date in brackets, etc. The authors are missing from ref 52 and 54 

Minor issues: tables currently run over multiple pages, ensure formatting to have them on single pages.

Page 3: Figure 1: Is the 'Scheme 2021' info below the reference for the figure? Where does the figure come from and do the authors have permission to reproduce it? Should also consider having it bigger, it is impossible to read the text as it is currently presented.

Page 4, Table 2, row 6 (in Disability Domains): 'neurodevelopmental disabilities effecting...' should be affecting

Page 5, line 157: This sentence is very unclear, please check the grammar: 'Action at the international level by ensuring that in the WHO STEPS data includes people with disabilities critically needed.'

Reviewer 4 Report

Nice paper overall and needed to continue the push for more PA for PWD. Here is my feedback.

Table1

- Key question for data quality row is confusing and tough for the reader to follow. Consider taking out the “don’t know” and state something like what percent of data is incomplete on ambiguous on the survey.

- Physical activity example in representative row. Maybe make that less wordy.

- Definition in timeliness row. …delay between processes…. The word steps there can get confusing since you’re talking about physical activity in the paper. The first instinct for me was that you meant steps as in walking. It took me that second to see steps as in processes or procedures.

Nice job with the Current Status of global physical activity surveillance among PWD. Very thorough and comprehensive yet easy to follow.

For table 2, have you considered adding specific studies even though they may not be governmental or institutional data collection methods? This section has so few data and I know there are other individual studies that examined rates in various areas of adults or children.

I’ll leave the decision to the editor, but I’m not sure if Table 3 is really needed. Some of this was explained and the source was given in the text so if the reader wanted this much detail, they can go to the Fulton et al. source.

Persons with SCI gaps – also are you looking at activity vs exercise? Vigorous intensity PA could be achieved with daily activities (ADLs) in someone with new SCI or older in age compared to someone that is younger and has adapted to the SCI. So, these factors play a big part in the gap of what the PA really is for this population.

Persons with stroke – line 197 you should include references to improvements in cardiovascular health as well as that’s a big reason why PA in PWD is important.

People with CP. Nice job on lines 258/259 with how habits set them up for adulthood activity with the disability.

Why only explore those 3 disabilities? What about TBI, Down Syndrome (DS), etc.? If there are reasons why other disabilities weren’t discussed, you should have a paragraph or two explaining those reasons. DS has had a lot of research about PA so that is glaring to me that you don’t mention it or the gaps that still exist in the literature around it.

Nice conclusion info.

Round 2

Reviewer 3 Report

Thank you for comprehensively addressing the reviewers comments, I have nothing further to add.